# Q-SLAM: Quadric Representations for Monocular SLAM

Chensheng Peng[1] *     Chenfeng Xu[1] *     Yue Wang[2]     Mingyu Ding[1]     Heng Yang[2]
Masayoshi Tomizuka[1]     Kurt Keutzer[1]     Marco Pavone[2]     Wei Zhan[1]

[1] UC Berkeley          [2] NVIDIA

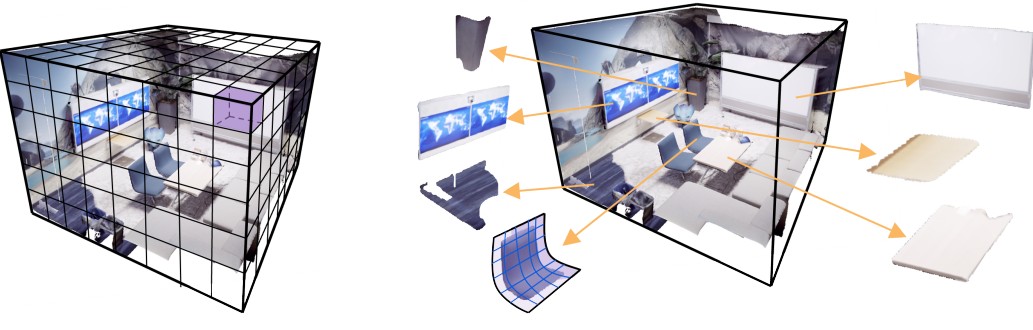

(a) Volumetric representation          (b) Quadric representation

Figure 1: Comparison between the volume representation and our quadric representation. We partition the 3D scenes into different quadric surfaces. Instead of processing every points independently, we can fully exploit the correlation between points on the same quadric surfaces.

**Abstract:** In this paper, we reimagine volumetric representations through the lens of quadrics. We posit that rigid scene components can be effectively decomposed into quadric surfaces. Leveraging this assumption, we reshape the volumetric representations with million of cubes by several quadric planes, which results in more accurate and efficient modeling of 3D scenes in SLAM contexts. First, we use the quadric assumption to rectify noisy depth estimations from RGB inputs. This step significantly improves depth estimation accuracy, and allows us to efficiently sample ray points around quadric planes instead of the entire volume space in previous NeRF-SLAM systems. Second, we introduce a novel quadric-decomposed transformer to aggregate information across quadrics. The quadric semantics are not only explicitly used for depth correction and scene decomposition, but also serve as an implicit supervision signal for the mapping network. Through rigorous experimental evaluation, our method exhibits superior performance over other approaches relying on estimated depth, and achieves comparable accuracy to methods utilizing ground truth depth on both synthetic and real-world datasets.

**Keywords:** Neural Radiance Fields, Simultaneous Localization and Mapping

## 1 Introduction

Recent years have seen a renaissance of simultaneous localization and mapping (SLAM), thanks to the advances in learning-based localization methods [1, 2, 3] and neural radiance fields (NeRFs) [4, 5, 6]. In its basic incarnation, SLAM involves estimating per-frame camera poses and reconstructing 3D scenes from visual inputs. The key challenging of monocular SLAM lies in the accurate 3D scene geometry modeling from visual inputs.

---

*Equal contribution.

8th Conference on Robot Learning (CoRL 2024), Munich, Germany.

To enable high-quality 3D scene reconstruction, recent SLAM approaches resort to NeRFs and Gaussians as key map representations. Specifically, NeRFs [7], employing neural networks *i.e.* MLPs, are recognized for their ability to achieve photo-realistic novel-view synthesis, as demonstrated in studies such as iMAP [5], Nice-SLAM [8]. Conversely, 3D Gaussian Splatting (3DGS) [9] relies on dense Gaussian representations to learn scene geometries effectively. Despite achieving photo-realistic novel-view synthesis, NeRFs and Gaussians have a few drawbacks when applied in SLAM systems. On the one hand, MLP-based NeRFs are slow to train while achieving significant memory efficiency. On the other hand, 3D Gaussian Splatting offers near-real-time training capabilities but suffer from large memory consumption and sensitivity to inaccurate camera pose estimation. Additionally, both methods focus on dense high-quality reconstruction, demanding substantial representation efforts to capture fine-grained details.

Addressing these challenges necessitates the design of map representations that are compatible with both learning-based localization front-end and NeRF(GS)-based reconstruction module. Therefore, to enhance scene geometry representation in SLAM, we introduce a novel method, termed **Q-SLAM**. As illustrated in Figure 1, we integrate the proposed quadric representations across the SLAM pipeline in several key areas.

**(1)** The well-established front-end tracking module can take RGB stream as input and offer a rough depth estimation while estimating camera poses. Nevertheless, this depth estimation encounters significant challenges, particularly at edges and in texture-less regions, where accuracy tends to degrade. To address these issues, we introduce a *quadric- based depth correction* module, leveraging quadric surfaces to refine the depth estimation, thus improving overall scene reconstruction accuracy, especially in complex and challenging environments.

**(2)** In the mapping module, departing from previous NeRF-based SLAM approaches like [4, 8], which rely on volumetric representations, our method transforms densely volumetric scenes - comprising millions of cubes (*e.g.*, $300 \times 300 \times 300$ voxels) - into a more manageable set of quadric surfaces (50-100 quadric surfaces per keyframe). This approach maintains the integrity of the 3D scene descriptions while significantly reducing complexity, by sampling around the quadric surfaces instead of the whole 3D space.

**(3)** During the rendering phase, we develop a *quadric-ray transformer*. We employ importance-sampling based on the quadric-rectified depth values, and we emphasize that sampled points on a quadric naturally belong to the same instance, which facilitates the modeling of their interrelationships during rendering, akin to the method in [10, 11]. We first perform feature interaction between points along a ray to aggregate information and then model the relationships between these surface points across rays using a transformer.

**(4)** Notably, we extend the utility of quadric semantics beyond mere depth correction and scene decomposition. They are also used as a supervision signal throughout the optimization of the NeRF network to learn robust and accurate features. With the integration of quadrics into the pipeline, we also consider the estimated pose as a learnable parameter in our mapping phase. This introduces a differentiable optimization process to refine the pose estimation, enhancing overall system accuracy.

## 2 Related Work

### 2.1 Dense Visual SLAM

Neural Radiance Fields (NeRF) [12] have exhibited notable efficacy in diverse applications, including view synthesis [12, 10, 13] and 3D reconstruction [14, 15]. **RGBD SLAM:** Innovative contributions such as iMAP [5] and Nice-SLAM [8] have pioneered the integration of NeRF into SLAM systems, demonstrating commendable tracking and mapping performance for indoor scenes. Co-SLAM [1] adopts a multi-resolution hash-grid representation for the scene to expedite convergence. E-SLAM [16] integrates the multi-scale axis-aligned perpendicular feature planes for efficient reconstruction of 3D scenes. By anchoring neural scene features in a point cloud generated iteratively in an input-driven manner, Point-SLAM [17] optimizes runtime and memory usage while maintain-

ing fine detail resolution. Besides using NeRF as the map representation, recent SLAM systems also resort to 3D Gaussian Splatting (3DGS) [9] due to its explicit representation property and real-time rendering performance. SplaTAM [18] pioneers in replacing NeRF with 3DGS and has achieved promising results in both tracking and mapping. GS-SLAM [19] further proposes a coarse-to-fine technique to select reliable 3D Gaussian representations for camera pose optimization effectively. However, these GS-based approaches have one drawbacks in common, much higher requirements for memory usage. All these approaches above rely on ground truth depth map for supervision of NeRF (or 3DGS) training, which is hard to obtain in real-world environment, thereby limiting their applicability in robotics. Conversely, **Monocular SLAM** method emerges, with only RGB images as input. In particular, NeRF-SLAM [4] integrates DROID-SLAM [20] for pose estimation and depth prediction and use Instant-NGP [21] to fit a NeRF. GO-SLAM [2] globally optimize poses and 3D reconstruction. However, the performance of these methods in 3D reconstruction is impeded by the inherent limitations of noisy predicted depth maps.

## 2.2 Scene representations

**Volumetric Representations** are predominantly utilized in Neural Radiance Fields (NeRFs) [22, 23, 12], and in 3D perception [24, 25, 26]. These representations encapsulate a 3D scene through a volumetric approach, where each volume element embodies occupancy probabilities or task-specific features. However, deriving volumetric representations solely from visual inputs presents significant challenges, particularly in terms of geometric estimation, leading to ambiguous and noisy outputs. **Point-Cloud Representation** offers a sparser alternative [27]. Point-SLAM [17] utilize a dynamic point density strategy to reduce computational and memory usage. Yet sparse point-clouds struggle to depict scenes comprehensively, while dense point-clouds encounter similar complications to volumetric representations, as previously mentioned. Consequently, few vision-based SLAM systems adopt point-based representations. Additionally, primitives like **Planes** [28] and **Quadric-Meshes** [29, 30, 31] have been proposed for scene representation. QuadricSLAM [32] estimates 3D quadric surfaces from 2D multi-view images and uses them as landmark representations for camera tracking in the SLAM pipeline. Dense Planar SLAM [33] uses bounded planes and surfels extracted from depth images to build a real-time SLAM system, but ignore the fine-grained reconstruction of 3D scenes. Point-plane SLAM [34] exploits the constraints of plane edges, which are predominant features that are less affected by measurement noise. ManhattanSLAM [35] utilizes the indoor structural information to estimate camera poses based on the Manhattan World (MW) assumption. However, these are typically employed for regularization during optimization or are challenging to optimize, limiting their applicability in visual SLAM. Drawing inspiration from **quadric representations** in LiDAR SLAM [3], we introduce quadric representations in visual monocular SLAM.

# 3 Q-SLAM

## 3.1 Overview

Our framework, Q-SLAM, as illustrated in Fig. 2, takes monocular RGB sequences as input. Initially, Droid-SLAM [20] predicts rough depth maps and initial camera poses from these inputs. Concurrently, a pretrained segmentation network is employed to estimate segmentation masks from these images. These masks are then utilized by the quadric-based depth correction module to refine the noisy depth maps, yielding more accurate corrected depth maps. Alongside the corresponding RGB images, camera poses, and segmentation results, they are inputted into the NeRF network.

For NeRF optimization, the RGB images, corrected depth and segmentation masks serve as supervision signals. To capture the semantic relationship across quadric surfaces, we further propose a quadric ray transformer, enabling effective feature interaction within and across the sampled rays. During the mapping process, both the camera poses $\mathbf{G} = g_t$ and the 3D scene representation parameters $\Theta$ are jointly optimized to enhance tracking and mapping accuracy. With the learned NeRF parameters, we can render RGB images, depth maps, and semantic maps for novel views, which can then be utilized to reconstruct the 3D mesh using TSDF-Fusion [36].

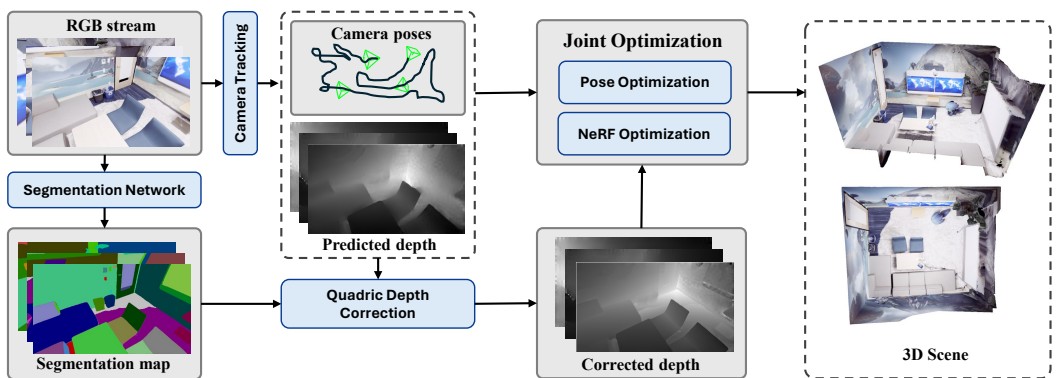

Figure 2: **Overview of our proposed method.** From the input RGB sequences, we can predict depth map, camera pose and segmentation mask. Subsequently, the initially estimated depth undergoes correction based on the quadric assumption. Along with the segmentation mask, camera poses, and image frames, the corrected depth are used for optimization of NeRF network. During the 3D reconstruction process, our proposed quadric ray transformer leverages the quadric priors effectively.

## 3.2 Quadric depth correction

Through the inverse re-projection with the pixel locations $(u, v)$ and the depth values $d$, a set of 3D points $p$ can be obtained for each segmented patch, where the depth value directly serves as the $z$ coordinate and $K$ is the calibration matrix: $[X, Y, Z]^T = d \cdot (K^{-1}[u, v, 1]^T)$

To represent the points of each segmented patch as quadric surfaces, we define the quadric implicit function as follows:

$$f(\mathcal{C}, \mathbf{x}) = \mathcal{C}_q^T \mathbf{q} + \mathcal{C}_l^T \mathbf{x} = c \tag{1}$$

where $\mathbf{x} = [x, y, z]^T$ is the linear term and $\mathbf{q} = [x^2, y^2, z^2, xy, yz, xz]^T$ is the quadric term calculated from $\mathbf{x}$, and $\mathcal{C}_q$, $\mathcal{C}_l$ and $c$ are the coefficients to be fitted. Following Narunas *et al.* [37], we define the cost function for the least-square fitting as the sum of squared distances between the quadric surface and the actual $N$ points to be fitted:

$$\mathbb{C} \triangleq \sum_{i=1}^{N} (\mathcal{C}_q \cdot \mathbf{q}_i + \mathcal{C}_l \cdot \mathbf{x}_i - c)^2 \tag{2}$$

Minimizing over Eq. 2, we obtain the quadric coefficients. We preserve the patches with fitting error below a predefined threshold, which implies a good fitting surface for the following depth correction. By substituting $(X, Y)$ back into $f(\mathcal{C}, \mathbf{x})$, an equation involving $z$ with other variables as constants is obtained. Solving this equation yields new $Z$ values, representing the rectified depth values.

The established quadric surface model proves effective in mitigating noise impact on depth information, ultimately improving the fidelity of the reconstructed spatial data. Fig. 3 illustrates the effectiveness of our proposed quadratically rectified approach. The original patch exhibits drifting points, particularly along the boundary where significant depth value changes occur. However, the proposed approach rectifies these points, fitting them to the quadric surface.

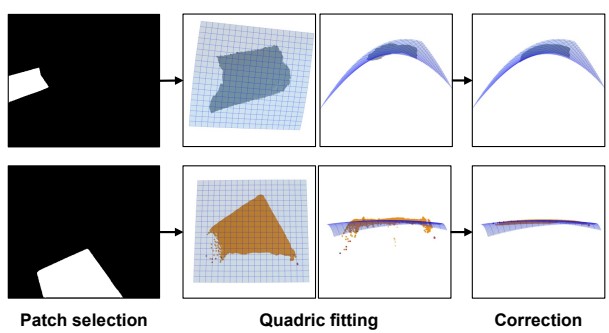

Patch selection     Quadric fitting     Correction

Figure 3: Quadric depth correction

The incorporation of additional constraints imposed by quadric surfaces contributes to a more precise depth prediction, thereby enhancing tracking and mapping accuracy.

### 3.3 Quadric Ray transformer

**Intra-Ray Transformer:** Drawing inspiration from IBRNet [10], our approach employs an attention mechanism along the ray to model relationships between sampled points. (Fig. 4)

Initially, $N_s$ points are uniformly sampled along each ray within the range $[d_{near}, d_{far}]$. Utilizing the corrected depth values, denoted as $d$, we refine the sampling process by selecting additional $N_d$ samples within the range of $[0.95d, 1.05d]$. Given the sampled points $p \in R^{B \times (N_s + N_d) \times 3}$, where $B$ is the number of sampled rays, we query the volume density features $f_\sigma \in R^{B \times (N_s + N_d) \times D}$ and feed them into the intra-ray transformer. Processing density features instead of color features is a wise choice as they solely depend on $xyz$ positions, simplifying feature aggregation based on the attention mechanism. Additionally, the smaller number of feature channels in density does not significantly increase the computational burden, resulting in an efficient representation for the application of SLAM.

The density features of each point along the ray are updated using self-attention:

$$f_\sigma' = \text{Attention}\left(Q(f_\sigma + \delta_p), K(f_\sigma + \delta_p), V(f_\sigma + \delta_p)\right) \tag{3}$$

where $\delta_p$ is the positional encoding in attention mechanism. This self-attention mechanism facilitates feature aggregation along a ray, enabling the capture of more information from the surface and nearby points for more precise surface reconstruction.

**Inter-Ray Transformer:** Intuitively, points belonging to the same semantic quadric are more likely to exhibit similar textural and spatial features, while rays sampled from different quadric surfaces might differ from each other in terms of textures and geometries. Therefore, we further propose the inter-ray transformer to capture the relationship across rays.

The inter-ray transformer operates on the updated density features from intra-ray transformer. First, the updated density features $f_\sigma'$ are concatenated with the semantic features $f_s$. Followed by a fusion network, new density features with semantic priors, $f_\sigma'' \in R^{B \times N \times D}$, can be obtained.

$$f_\sigma'' = \text{MLP}_{fusion}\left(f_\sigma' \oplus f_s\right) \tag{4}$$

As shown in Fig. 4, the inter-ray transformer then takes the transposed density features, $f_\sigma''^T \in R^{N \times B \times D}$, as input, where attention map is calculated across different rays along the $B$ dimension:

$$f_\sigma'''^T = \text{Attention}\left(Q(f_\sigma''^T + \delta_p'), K(f_\sigma''^T + \delta_p'), V(f_\sigma''^T + \delta_p')\right) \tag{5}$$

where $\delta_p'$ is the positional encoding in attention mechanism.

The inter-ray transformer further facilitates feature interaction across rays. This process allows surface points to capture more inter-ray information from other rays from the same quadric surface. Together with the intra-ray transformer, our proposed network achieves feature aggregation across a broad range of ray points, enhancing rendering accuracy by incorporating quadric priors and additional spatial information.

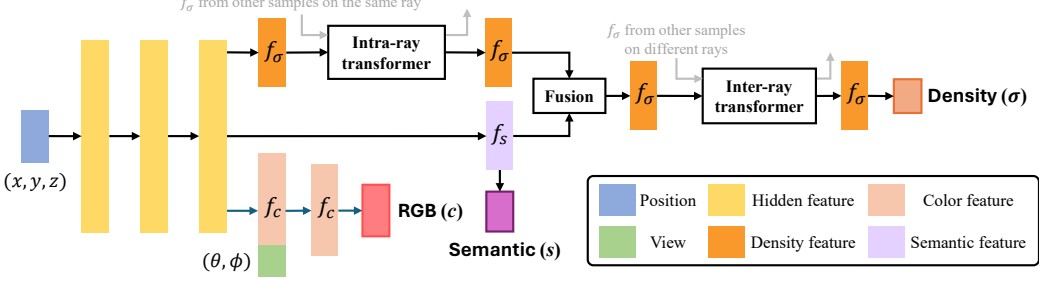

Figure 4: The detailed structure of quadric ray transformer.

### 3.4 Joint Optimization of Rendering and Pose Estimation

During mapping, we map the 3D coordinates $\mathbf{x}$, and viewing directions $\mathbf{d}$ to volume density, color and semantic logits $(\sigma, \mathbf{c}, \mathbf{s})$. Subsequently, volume rendering is utilized to reconstruct color $\hat{\mathbf{C}}$, depth $\hat{\mathbf{D}}$, and semantic map $\hat{\mathbf{S}}$. For points $\{p_i | i = 1, \cdots, M\}$ on a fitted quadric surface, the color and depth loss is calculated as follows:

$$\mathcal{L}_c = \frac{1}{\varepsilon M} \sum_{m=1}^{M} \left| \mathbf{C}_m - \hat{\mathbf{C}}_m \right|; \mathcal{L}_d = \frac{1}{\varepsilon M} \sum_{m=1}^{M} \left| \mathbf{D}_m - \hat{\mathbf{D}}_m \right| \tag{6}$$

Here, the fitting error $\varepsilon$ serves as an uncertainty penalty. In cases where the fitting is not perfect, the quadric plane is less likely to exhibit similar texture and spatial features, resulting in reduced usefulness of the quadric transformer. Consequently, such patches are down-weighted by incorporating the fitting error $\varepsilon$. Following [38], we use a multi-class cross-entropy loss as the semantic loss $\mathcal{L}_s$. Hence the rendering loss is a weighted sum of $\mathcal{L}_c$, $\mathcal{L}_d$ and $\mathcal{L}_s$ with hyperparameters $\lambda_1, \lambda_2$.

$$\mathcal{L} = \mathcal{L}_c + \lambda_1 \mathcal{L}_d + \lambda_2 \mathcal{L}_s \tag{7}$$

For joint optimization, the camera pose $\mathbf{G} = \{g_t\}$, a $4 \times 4$ transformation matrix, is converted to quaternion ($4 \times 1$) and translation ($3 \times 1$) vector, which are then taken as trainable parameters to the network, optimized together with the NeRF parameters while minimizing the rendering loss. In this way, the camera poses $\mathbf{G}$ and network parameters $\Theta$ can be jointly optimized.

## 4 Experiments

### 4.1 Experimental Setup

**Dataset.** Q-SLAM is evaluated on a variety of datasets, including Replica [39], ScanNet [40], and KITTI MOT dataset [41]. For evaluation of the reconstruction quality, we test our method on 8 synthetic scenes from Replica, which provides high-quality synthetic scenes. Following GO-SLAM [2], we evaluate the tracking accuracy on ScanNet dataset which offers extensively annotated RGB-D scans of real-world scenarios, encompassing challenging short and long trajectories. KITTI MOT dataset is an outdoor autonomous driving dataset with dynamic objects, such as cars. For camera tracking assessment, our approach is evaluated under two distinct modes: one utilizing ground truth and the other utilizing estimated depth from monocular images as inputs.

**Metrics.** We evaluate tracking accuracy by aligning the estimated trajectory with the ground truth trajectory and computing the Root Mean Square Error (RMSE) of the Absolute Trajectory Error (ATE). This metric quantifies the Euclidean distance between the estimated pose and the corresponding ground truth pose. Following NeRF-SLAM [4], we utilize Peak Signal-to-Noise Ratio (PSNR), SSIM [42], and LPIPS [43] for image rendering evaluation, which evaluate the similarity between the rendered images and the ground truth images, and Accuracy [cm], Completion [cm], Completion Ratio [%] for 3D reconstruction assessment, which measures the reconstruction quality by comparing the reconstructed mesh and the ground truth mesh.

Table 1: Photometric (PSNR [dB], SSIM, LPIPS) and Geomertric (Acc. [cm], Comp. [cm], Comp. Ratio [%]) results on Replica dataset [39].

| Setting | Method | PSNR ↑ | SSIM ↑ | LPIPS ↓ | ATE ↓ | Acc. ↓ | Comp. ↓ | Comp. Ratio (%) ↑ |
|---------|--------|--------|--------|---------|-------|--------|---------|-------------------|
| RGBD | Nice-SLAM [8] | 24.42 | 0.81 | 0.23 | 1.95 | 3.87 | 3.87 | 82.41 |
| | Vox-Fusion [44] | 24.41 | 0.80 | 0.24 | 0.54 | 2.67 | 4.55 | 86.59 |
| | SplaTAM [18] | 34.11 | 0.97 | **0.10** | 0.36 | 2.88 | 3.57 | 71.68 |
| | Ours | **35.34** | **0.98** | 0.12 | **0.34** | **2.17** | **2.47** | **91.13** |
| Mono | COLMAP [45] | 13.94 | 0.72 | 0.35 | - | 8.69 | 12.12 | 67.62 |
| | DROID-SLAM [20] | 21.69 | 0.81 | 0.25 | 0.42 | 5.50 | 12.29 | 63.62 |
| | Nicer-SLAM [46] | 25.41 | 0.83 | 0.19 | 1.88 | 3.65 | 4.16 | 79.37 |
| | Ours | **32.49** | **0.89** | **0.17** | 0.38 | **2.89** | **3.55** | **84.79** |

Table 2: ATE RMSE [cm] Results on ScanNet dataset [40]. 'VO' denotes visual odometry. Results of DROID-SLAM are from [2] and results of iMAP* and Nice-SLAM are from [8].

| Setting | Scene ID | 0000 | 0054 | 0233 | 0465 | 0059 | 0106 | 0169 | 0181 | Avg. |
|---|---|---|---|---|---|---|---|---|---|---|
|  | # Frames | 5578 | 6629 | 7643 | 6306 | 1807 | 2324 | 2034 | 2349 |  |
| RGBD | iMAP* [5] | 55.95 | 70.11 | 86.42 | 85.03 | 32.06 | 17.50 | 70.51 | 32.10 | 56.21 |
|  | Nice-SLAM [8] | 8.64 | 20.93 | 9.00 | 22.31 | 12.25 | 8.09 | 10.28 | 12.93 | 13.05 |
|  | DROID-SLAM (VO) [20] | 8.00 | 29.28 | 6.75 | 11.37 | 11.30 | 9.97 | 8.64 | 7.38 | 11.59 |
|  | DROID-SLAM [20] | 5.36 | 8.89 | 4.90 | 8.32 | 7.72 | 7.06 | 8.01 | 6.97 | 7.15 |
|  | GO-SLAM [2] | 5.35 | 8.75 | 4.78 | **8.15** | **7.52** | 7.03 | 7.74 | 6.84 | 7.02 |
|  | Ours | **5.23** | **8.57** | **4.68** | 8.33 | 7.63 | **7.02** | **7.66** | **6.52** | **6.96** |
| Mono | DROID-SLAM (VO) [20] | 11.05 | 204.31 | 71.08 | 117.84 | 67.26 | 11.20 | 16.21 | 9.94 | 63.61 |
|  | DROID-SLAM [20] | 5.48 | 197.71 | 72.23 | 114.36 | 9.00 | **6.76** | **7.86** | **7.41** | 52.60 |
|  | GO-SLAM [2] | 5.94 | 13.29 | 5.31 | 79.51 | **8.27** | 8.07 | 8.42 | 8.29 | 17.59 |
|  | Ours | **5.77** | **12.62** | **5.27** | **76.96** | 8.46 | 8.38 | 8.74 | 8.76 | **16.87** |

**Implementation Details.** All experiments are conducted on an NVIDIA A6000 GPU using PyTorch 1.10.0. We use Adam as the optimizer with $\beta_1 = 0.9$ and $\beta_2 = 0.999$. The tracking module is adapted from Droid-SLAM [20], and pretrained weights are utilized to estimate depths and poses. Our mapping backbone is adapted from Point-SLAM [17], incorporating our proposed depth correction and quadric ray transformer. For faster image segmentation, we use the MobileSAM [47] to obtain segmentation masks from each input RGB frame. Diverging from methodologies where scene meshes are reconstructed using marching cubes on Signed Distance Function (SDF) values of queried points, Q-SLAM renders images and depths over the estimated camera trajectory and use TSDF-Fusion [36] for mesh construction with voxel size 1 cm. During the fitting process, only quadric surfaces with fitting error lower than the specified threshold undergo depth correction; otherwise, the uncorrected predicted depth is used for the supervision of NeRF training. For joint optimization, the scene representation parameters $\Theta$ are optimized for five steps, and the accumulated losses are then utilized to update camera pose $\mathbf{G}$.

## 4.2 Comparison with SOTA

**Replica dataset.** We evaluate on the 8 scenes as Nice-SLAM and iMAP. As shown in Tab. 1, the results demonstrate comparable performance across almost all scenes in terms of rendering and reconstruction metrics. Notably, our approach exhibits performance on par with SLAM systems that utilize ground truth depth for supervision.

**ScanNet dataset.** For a comprehensive comparison, we further evaluate our SLAM system on ScanNet [40] dataset, encompassing both long and short sequences. As depicted in Tab. 2, our method outperforms other approaches in both monocular and RGBD setting, with a more pronounced superiority evident in the case of monocular inputs.

**Runtime and Memory Usage.** In Table 3, we also report the runtime and memory usage on the Replica dataset. The tracking and mapping time is reported per frame. It can be observed that our method can achieve comparable speed with Vox-Fusion [44], but providing much higher rendering quality as shown in Tab 1. We also report the runtime of individual breakdown components in Tab 5.

Table 3: Runtime and memory usage.

| Method | Memory | Tracking | Mapping |
|---|---|---|---|
| Nice-SLAM [8] | 10.13 GB | 1.32s | 10.92s |
| Vox-Fusion [44] | 12.45 GB | 0.36s | 0.55s |
| Ours | 14.59 GB | 0.44s | 0.89s |

## 4.3 Ablation Study

**Depth Correction.** In Tab. 4 (a), we compare the performance with or without the proposed quadric depth correction module. It can be observed that the depth correction process enhances the accuracy of the predicted depth map by introducing additional surface constraints, rectifying drifting points that deviate from the surfaces. This improvement positively impacts both tracking and mapping performance, especially for the Depth L1 metric.

Table 4: Ablation on Replica dataset.

| Setting | Depth L1 | PSNR | ATE |
|---|---|---|---|
| Full Setting | **2.76** | **32.49** | **0.38** |
| (a) w/o depth correction | 3.23 | 31.42 | 0.40 |
| (b) w/o quadric transformer | 2.98 | 31.87 | 0.39 |
| (c) w/o joint optimization | 2.81 | 32.14 | 0.42 |
| (d) w/o semantic supervision | 2.91 | 32.09 | 0.39 |

Table 5: Breakdown of runtime.

| Components | Timing (per frame) |
|---|---|
| Droid-SLAM prediction | 54 ms |
| MobileSAM segmentation | 37 ms |
| Depth Correction | 143 ms |
| Tracking (pose optimization) | 442 ms |
| Mapping | 894 ms |

| Co-SLAM | GO-SLAM | Ours | Ground Truth |
|---|---|---|---|

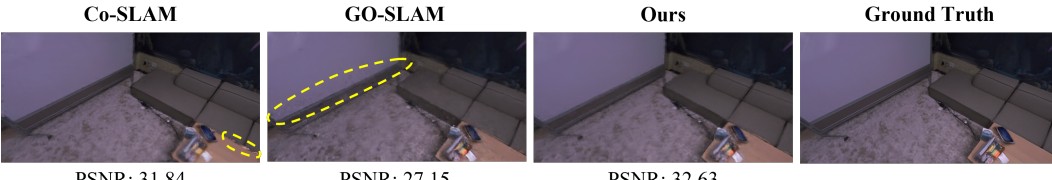

| PSNR: 31.84 | PSNR: 27.15 | PSNR: 32.63 | |

Figure 5: Qualitative reconstruction results on Replica dataset. We compare our solution with recent SOTA SLAM systems Co-SLAM [1] and GO-SLAM [2]. Our method can recover better texture features, especially on the boundary of instances.

**Quadric Transformer.** In experiments without the quadric ray transformer, the points along a sampled ray are processed independently, lacking feature interaction between each other. The introduction of the quadric ray transformer can help capture information along the ray, and hence improve the performance, as illustrated in Tab. 4 (b).

**Joint Optimization.** For experiments without joint optimization, the estimated camera poses are not optimized using the rendering loss during the mapping process. It can be observed from Tab. 4 (c) that joint optimization of camera poses and 3D reconstruction demonstrates a modest improvement in both tracking and mapping accuracy. We believe that the limited improvement stems from the fact that joint optimization is solely conducted on keyframes during the mapping process. Given that keyframes constitute only a small portion of the entire sequence, while the evaluation of camera poses spans all frames, this potentially constrains the impact of joint optimization.

**Semantic Supervision.** With an additional semantic head, our model not only renders color and depth but also generates the semantic map. It's intuitive to utilize segmentation results from previous modules as a supervisory signal, enriching the available information. As shown in Table 4 (d), incorporating semantic supervision can significantly enhance performance.

## 5 Conclusion

In this paper, we introduce quadratic surfaces as the map representation for SLAM. Based on segmentation results from RGB images, the roughly estimated depth values can be corrected by incorporating additional surface constraints. We utilize the rectified depths and quadric semantics as a prior for sampling points along the ray, significantly reducing the required number of samples to achieve comparable results, thus alleviating the computational burden. Additionally, we employ a novel quadric-ray transformer model to capture interrelations across different samples along the ray within the constraints of quadric surfaces. Furthermore, we propose an end-to-end joint optimization approach for pose estimation and 3D reconstruction. Sufficient experiments on various datasets demonstrate the effectiveness of our proposed method in terms of novel-view synthesis, depth estimation, and camera tracking.

**Limitations.** The heavy computational cost of our method compared to traditional SLAM systems and the difficulty in scaling to unbounded outdoor scenarios. We also acknowledge that the quadric assumption of the surface is not ubiquitous, especially for complex structures in outdoor scenarios, such as trees. Additionally, the smoothing process heavily relies on the segmentation results. We plan to explore more efficient representations, such as gaussian splatting, in the future work.

**Acknowledgments**

This work is supported by Berkeley DeepDrive. [2]

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

# Appendix / Supplemental Materials

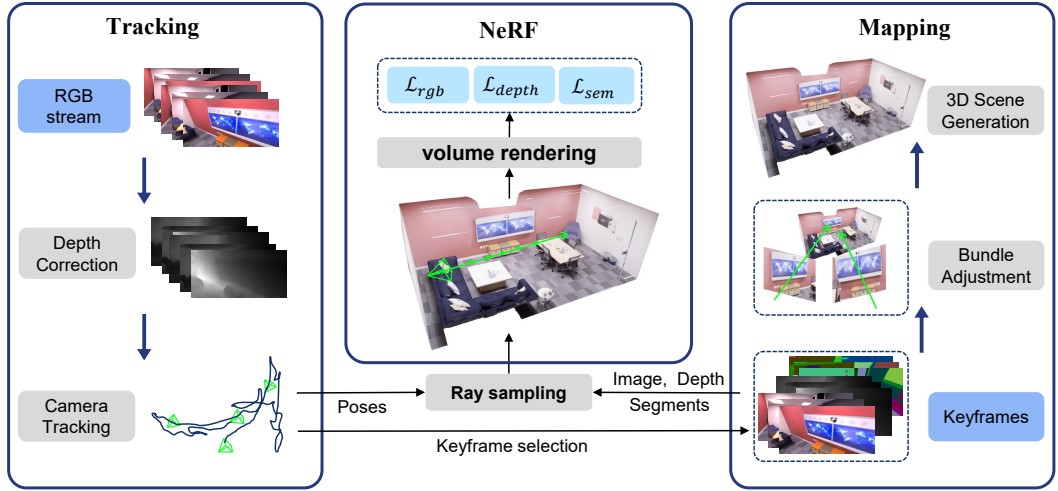

Figure 6: Structure overview of Q-SLAM. 1) Tracking: initialize per-frame camera poses and depth prediction. Correct the noisy depth using our proposed depth correction module based on the segmentation results from monocular inputs. 2) NeRF: using the selected keyframes to supervise the optimization of NeRF network equipped with our proposed quadric-decomposed transformer. 3) Mapping: global bundle adjustment to jointly optimize the scene representation and camera poses taking rays sampled from all keyframes. Reconstruct the complete scene by fusing the rendered RGB images and depth maps with TSDF-fusion [36].

## A  Additional Methodology Details

### A.1  Camera tracking

The initial depth and camera poses are obtained from the learning-based Droid-SLAM [20]. Taking sequences of RGB images as input, Droid-SLAM can predict camera pose and pixel-wise depth through recurrent iterative updates. We do not make changes to the Driod-SLAM codebase but use them to initialize the predicted depths and poses. The rough depths are then rectified with our proposed depth correction module, and the camera poses are optimized with our joint optimization technique.

### A.2  Local bundle adjustment

By setting a window size of 25, we will perform a local bundle adjustment. Following Nice-SLAM [8], we treat camera poses as trainable parameters. We sample 400 rays per image within the window and optimize their corresponding poses using the rendering loss.

$$\mathcal{L} = \mathcal{L}_c + \lambda_1 \mathcal{L}_d + \lambda_2 \mathcal{L}_s \tag{8}$$

where $\mathcal{L}_c$ is the color loss, $\mathcal{L}_d$ is the depth loss and $\mathcal{L}_s$ is the semantic loss with hyperparameters $\lambda_1, \lambda_2$.

### A.3  Keyframe management

Following Droid-SLAM, our system takes as input a live RGB stream, and applies a recurrent update operator based on RAFT [48] to compute the optical flow of each new frame compared to the last keyframe. If the average flow is larger than a predefined threshold $\tau_{flow}$, a new keyframe is created out of the current frame and added to the maintained keyframe buffer.

## A.4 Quadirc surface fitting and depth correction

By setting $\nabla \mathbb{C}_c = 0$ in Eq. 2 to obtain the optimal $c^*$

$$c^* = \frac{1}{N} \sum_{i=1}^{N} (\mathcal{C}_q \cdot \mathbf{q}_i + \mathcal{C}_l \cdot \mathbf{x}_i) \triangleq \mathcal{C}_q \cdot \bar{\mathbf{q}} + \mathcal{C}_l \cdot \bar{\mathbf{x}} \tag{9}$$

The cost function in Eq. 2 becomes:

$$\mathbb{C} = \sum_{i=1}^{N} (\mathcal{C}_q \cdot (\mathbf{q}_i - \bar{\mathbf{q}}) + \mathcal{C}_l \cdot (\mathbf{x}_i - \bar{\mathbf{x}}))^2 \tag{10}$$

where $\bar{\mathbf{q}} = \frac{1}{N} \sum_{i=1}^{N} q_i$ are the quadric term averaged on points in a patch, $\bar{\mathbf{x}} = \frac{1}{N} \sum_{i=1}^{N} x_i$ is the linear term.

The intermediate variables are defined as follows:

$$\begin{aligned}
\mathbb{L} &\triangleq \sum_{i=1}^{N} (\mathbf{x}_i - \bar{\mathbf{x}})(\mathbf{x}_i - \bar{\mathbf{x}})^T \\
\mathbb{M} &\triangleq \sum_{i=1}^{N} (\mathbf{q}_i - \bar{\mathbf{q}})(\mathbf{q}_i - \bar{\mathbf{q}})^T \\
\mathbb{N} &\triangleq -\sum_{i=1}^{N} (\mathbf{q}_i - \bar{\mathbf{q}})(\mathbf{x}_i - \bar{\mathbf{x}})^T
\end{aligned} \tag{11}$$

Setting $\nabla \mathbb{C}_{\mathcal{C}_l} = 0$ gives

$$\mathbb{L}\, \mathcal{C}_l^* = \mathbb{N}^T \mathcal{C}_q \tag{12}$$

By substituting $\mathcal{C}_l^*$ back to Eq. 10, we can obtain

$$\begin{aligned}
\mathbb{C} &\triangleq \sum_{i=1}^{N} \left\| \left( (\mathbf{q}_i - \bar{\mathbf{q}})^T + (\mathbf{x}_i - \bar{\mathbf{x}})^T \mathbb{L}^{-1} \mathbb{N}^T \right) \mathcal{C}_q \right\|^2 \\
&= \mathcal{C}_q{}^T \Psi \mathcal{C}_q, \text{ where } \Psi \triangleq \mathbb{M} - \mathbb{N}\, \mathbb{L}^{-1} \mathbb{N}^T
\end{aligned} \tag{13}$$

Minimizing Eq. 13 over $c_q$ gives the eigenvector $c_q^*$ of $\Psi$ corresponding to the minimum eigenvalue, and $c_l^*$ can be solved from Eq. 12, and $c^*$ from Eq. 9.

As defined by Taubin *et al*. [49], the distance from a point $\mathbf{x}$ to a quadric surface $f$ is:

$$d(\mathbf{x}, f) \approx \frac{f^2(\mathcal{C}, \mathbf{x})}{|\nabla_{\mathbf{x}} f(\mathcal{C}, \mathbf{x})|^2} \tag{14}$$

For every fitted patch, we calculate the average distance between the original points to the fitted surface as the fitting error. Those patches with error exceeding the given threshold will be discarded. We only preserve the patches with relatively small fitting error, which implies a good fitting surface for the following depth correction.

## A.5 Ray points sampling

Initially, rays are constructed from the provided image and calibration matrix, and $N_s$ points are uniformly sampled along each ray within the range $[d_{near}, d_{far}]$. Utilizing the corrected depth values, denoted as $d$, we refine the sampling process by selecting additional $N_d$ samples within the range of $[0.95d, 1.05d]$. We do not simply sample around $d$ because potential errors in the corrected depth values might lead to an extended sampling distance away from the true surface.

## A.6 Extension to dynamic scenes

While the static assumption applied in existing NeRF SLAM methods is suitable for indoor environments, it becomes inadequate for outdoor scenarios with moving objects, particularly in autonomous

| GO-SLAM | Co-SLAM | Ours |
|---------|---------|------|

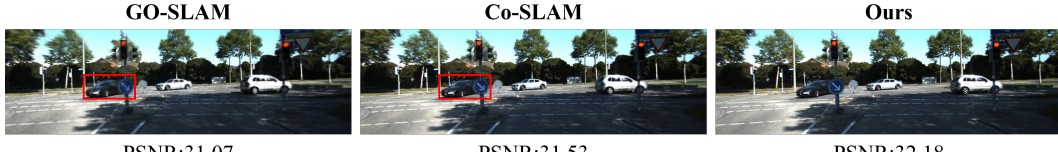

| PSNR:31.07 | PSNR:31.53 | PSNR:32.18 |
|------------|------------|------------|

Figure 7: Qualitative reconstruction results on KITTI MOT dataset. We compare our solution with recent SOTA NeRF-SLAM systems Co-SLAM [1] and GO-SLAM [2].

driving contexts, limiting their applicability. We propose to extend our method to outdoor scenarios, which are rarely explored in previous methods [8, 17, 16, 18]. By incorporating an additional dynamic branch, we can effectively model the reconstruction of moving objects using semantic priors. Unlike the static branch, which outputs $(\sigma_s, \boldsymbol{c}_s)$, the dynamic branch produces outputs $(\sigma_d, \boldsymbol{c}_d)$ that vary with time $t$. The time embedding, derived from $t$ is integrated into the dynamic branch along with other inputs. The final density $\sigma$ and color $\boldsymbol{c}$ for rendering are calculated as follows:

$$\sigma(\mathbf{x}, t) = \sigma_s(\mathbf{x}) + \sigma_d(\mathbf{x}, t), \quad \boldsymbol{c}(\mathbf{x}, \mathbf{d}, t) = \frac{\sigma_s}{\sigma}\boldsymbol{c}_s(\mathbf{x}, \mathbf{d}) + \frac{\sigma_d}{\sigma}\boldsymbol{c}_d(\mathbf{x}, \mathbf{d}, t) \tag{15}$$

where $\mathbf{x}$ and $\mathbf{d}$ represents the 3D coordinates and viewing direction respectively, and $t$ is the timestamp. The subscript $s$ and $d$ stand for static and dynamic branch respectively, and the outputs of dynamic branch depend on time, while the static branch does not.

The color $\hat{C}$ and depth $\hat{D}$ are rendered as follows:

$$\hat{C}(\mathbf{r}, t) = \int_0^{+\infty} T(s)\sigma(\mathbf{r}(s), t)\boldsymbol{c}(\mathbf{r}(s), \mathbf{d}, t)\, ds \tag{16}$$

$$\hat{D}(\mathbf{r}, t) = \int_0^{+\infty} T(s)\sigma(\mathbf{r}(s), t)ds \tag{17}$$

We do not incorporate the semantic head for outdoor scenes, because there are much more classes of objects compared to indoor scenes. To supervise the training on dynamic objects, we generate the static masks following [50], and apply a regularization loss of dynamic branch on the static regions.

$$\mathcal{L}_r = \sum_{\mathbf{x} \in \text{static}} |\sigma_d(\mathbf{x}, t)|_1 \tag{18}$$

We conducted experiments on KITTI dataset [41], including outdoor scenes with moving cars. The qualitative results are presented in Fig. 7.

## B  Additional Implementation Details

### B.1  Data source

The data of other NeRF-SLAM methods in Tab. 1 is sourced from Nicer-SLAM [46], and the geometric reconstruction results (Acc., Comp., etc.) of SplaTAM [18] comes from RTG-SLAM since the original paper does not report these metrics. The results of other method in Tab. 2 are mainly taken from GO-SLAM [2].

### B.2  Dataset

Q-SLAM is evaluated on a variety of datasets, including Replica [39], ScanNet [40], and TUM RGB-D [51] dataset. For evaluation of reconstruction quality, we test our method on 8 synthetic scenes from Replica, which provides high-quality synthetic scenes, akin to the evaluation framework adopted by NeRF-SLAM [4]. Following GO-SLAM [2], we evaluate the tracking accuracy

on ScanNet dataset which offers extensively annotated RGB-D scans of real-world scenarios, encompassing challenging short and long trajectories. Following Nice-SLAM [8], we also evaluate on various scenes on indoor TUM RGB-D dataset, with ground truth poses provided by a motion capture system. For camera tracking assessment, our approach is tested under two distinct modes: one utilizing ground truth and the other utilizing estimated depth from monocular images as inputs. The batch size of sampled rays to NeRF network is 8192.

## B.3 Evaluation Metrics

We evaluate tracking accuracy by aligning the estimated trajectory with the ground truth trajectory and computing the Root Mean Square Error (RMSE) of the Absolute Trajectory Error (ATE). This metric quantifies the Euclidean distance between the estimated pose and the corresponding ground truth pose. In line with the evaluation approach of NeRF-SLAM [4], we utilize Peak Signal-to-Noise Ratio (PSNR), SSIM [42], and LPIPS [43] for image rendering evaluation, and Accuracy [cm], Completion [cm], Completion Ratio [%] for 3D reconstruction assessment.

- Absolute Trajectory Error (ATE) (cm) ↓: Evaluates trajectory estimation accuracy by measuring the average Euclidean translation distance between corresponding poses in estimated and ground truth trajectories, often reported in terms of Root Mean Square Error (RMSE).

- Peak Signal to Noise Ratio (PSNR) ↑: Measures image quality by evaluating the ratio between the maximum pixel value and the root mean squared error, usually expressed in terms of the logarithmic decibel scale.

- Structural Similarity Index Measure (SSIM) ↑: Assesses image quality by examining the similarities in luminance, contrast, and structural information among patches of pixels.

- Learned Perceptual Image Patch Similarity (LPIPS) ↓: Utilizes learned convolutional features to assess image quality based on feature map mean squared error across layers.

- Accuracy (cm) ↓: Computes the average distance between sampled points from the reconstructed mesh and the nearest ground-truth point.

- Completion (cm) ↓: Measures the average distance between sampled points from the ground-truth mesh and the nearest reconstructed.

- Completion Ratio (%) ↑: the percentage of points in the reconstructed mesh with Completion under 5 cm.

## B.4 Hyperparameters

All experiments are conducted on NVIDIA A6000 GPU with PyTorch 1.10.0. We use Adam as our optimizer with $\beta_1 = 0.9, \beta_2 = 0.999$. The tracking backbone is Droid-SLAM, where we use the pretrained weights to estimate depths and poses. We use Point-SLAM as the mapping backbone, equipped with our proposed depth correction and quadric transformer. The threshold for motion filter is 4.0 pixels, a tracked frame is considered as a keyframe only if the average optical flow is greater than the threshold. The window size for local bundle adjustment is 25. During the joint optimization process, camera poses are optimized for one epoch, and the NeRF network parameters are optimized for five epochs.

## B.5 Segmentation and Quadric Fitting

For the segmentation network, we use Segment Anything Model (SAM) [52] and MobileSAM [47] for faster inference, an off-the-shelf network to produce the mask for quadric fitting. To prevent the negative effect of outliers, quadric fitting only applies to segments with area larger than 200 pixels.

During the fitting process, we calculate the coefficient of determination to evaluate the fitting performance. Let $z_i$ be the predicted depth value and $f_i$ be the corresponding corrected depth. The

coefficient of determination is calculated as follows:

$$\bar{z} = \frac{1}{n}\sum_{i=1}^{n} z_i$$

$$SS_{\mathrm{res}} = \sum_i (z_i - f_i)^2$$

$$SS_{\mathrm{tot}} = \sum_i (z_i - \bar{z})^2$$

$$R^2 = 1 - \frac{SS_{\mathrm{res}}}{SS_{\mathrm{tot}}}$$

We only perform depth correction on quadric surfaces with fitting coefficient greater than the given threshold 0.85, otherwise, we just use the predicted depth for the supervision of NeRF training.

### B.6   Mesh Reconstruction

Different from other approaches that reconstruct mesh of a scene by running marching cubes on the Signed Distance Function (SDF) values of the queried points, we render images and depths for the selected keyframes. The reason for the difference is that our rendering requires to correlate points along and across rays, while other approaches process 3D points independently. We first render RGB images and depth maps, and then use TSDF-fusion [36] to reconstruct the 3D volume mesh.

## C   Additional Experimental Results

**TUM-RGBD dataset.** We evaluate the tracking performance of our methods on the small-scale indoor-scene dataset with two different inputs, monocular and RGBD images. As presented in Table 6, our approach outperforms traditional SLAM, including ORB-SLAM2 [53] and ORB-SLAM3 [54], which exhibits failures in certain scenarios. In comparison to recent NeRF-based SLAM systems, our solution consistently achieves superior results across most scenes. We attribute the improvements to our proposed quadric representation and quadric transformer, especially for scenes with well-segmented planes and surfaces such as desks, floors, and rooms.

Following GO-SLAM [2], we also test our solution with RGBD images as input, as indicated in Table 7. While the quadric-based depth correction is not performed under this setting, our proposed quadric ray transformer and semantic supervision also contribute to the performance improvement.

## D   Visualization

In Fig. 8, we provide the qualitative results of the reconstruction. It can be observed that our method outperforms GO-SLAM, especially on the boundaries of objects.

Table 6: ATE RMSE [m] Results on TUM [51] dataset freiburg1 set (monocular setting). ORB-SLAM2 [53] and ORB-SLAM3 [54] fail on certain scenes.

| | 360 | desk | desk2 | floor | plant | room | rpy | teddy | xyz | avg |
|---|---|---|---|---|---|---|---|---|---|---|
| ORB-SLAM2 [53] | - | 0.071 | - | 0.023 | - | - | - | - | 0.010 | - |
| ORB-SLAM3 [54] | - | 0.017 | 0.210 | - | 0.034 | - | - | - | 0.009 | - |
| DeepV2D [55] | 0.243 | 0.166 | 0.379 | 1.653 | 0.203 | 0.246 | 0.105 | 0.316 | 0.064 | 0.375 |
| DeepFactors [56] | 0.159 | 0.170 | 0.253 | 0.169 | 0.305 | 0.364 | 0.043 | 0.601 | 0.035 | 0.233 |
| DROID-SLAM [20] | 0.111 | 0.018 | 0.042 | **0.021** | **0.016** | 0.049 | 0.026 | **0.048** | 0.012 | 0.038 |
| GO-SLAM[2] | 0.089 | 0.016 | 0.028 | 0.025 | 0.026 | 0.052 | **0.019** | **0.048** | 0.010 | 0.035 |
| Ours | **0.086** | **0.013** | **0.023** | 0.026 | 0.027 | **0.049** | 0.021 | 0.049 | **0.009** | **0.033** |

Table 7: ATE [m] Results on TUM dataset [51] with RGB-D inputs from freiburg1, freiburg2 and freiburg3 set.

| Method | fr1/desk | fr2/xyz | fr3/office |
|---|---|---|---|
| Kintinuous [57] | 0.037 | 0.029 | 0.030 |
| BAD-SLAM [58] | 0.017 | 0.011 | 0.017 |
| ORB-SLAM2 [53] | 0.016 | **0.004** | **0.010** |
| iMAP [5] | 0.049 | 0.020 | 0.058 |
| NICE-SLAM [8] | 0.027 | 0.018 | 0.030 |
| Ours | **0.014** | 0.005 | 0.011 |

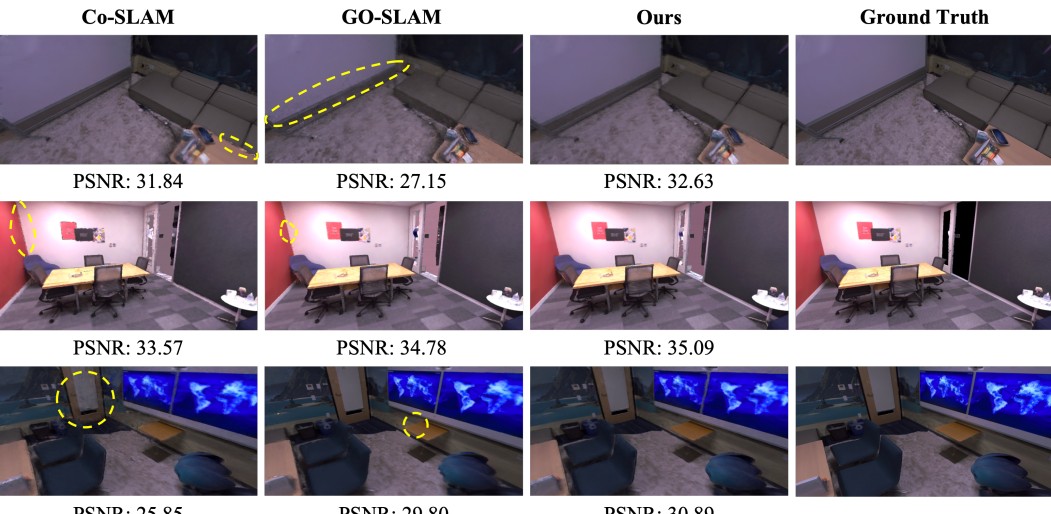

Figure 8: Qualitative reconstruction results on Replica dataset. We compare our solution with recent SOTA SLAM systems Co-SLAM [1] and GO-SLAM [2]. Our method can recover better texture features, especially on the boundary of instances.

We provide several selected visualization results for depth correction as shown in Fig. 9. Benefiting from the segmentation mask, the depth correction improves the sharpness of the boundary of objects.

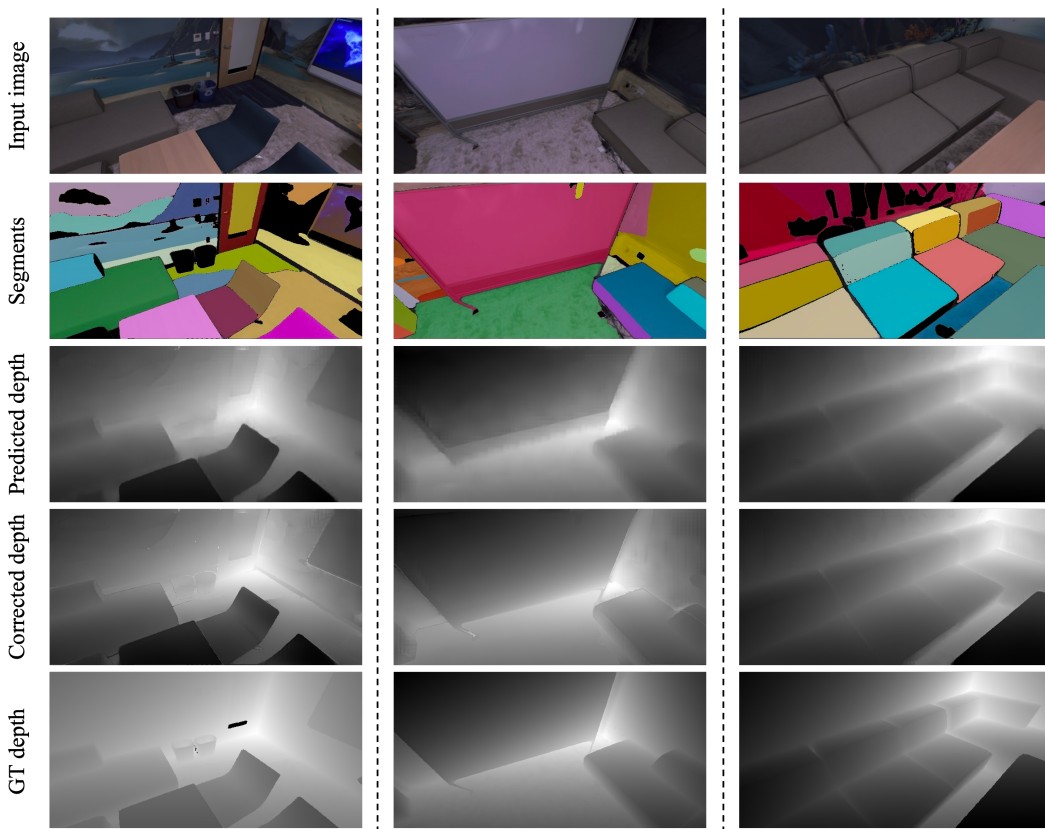

Figure 9: Qualitative results of depth correction.

