# OpenReview forum: "Q-SLAM: Quadric Representations for Monocular SLAM"
_robot-learning.org/CoRL/2024/Conference — CoRL 2024_

### Official Review · Reviewer_csiv · 2024-07-20
**The design of the SLAM system reads reasonable for scenes containing many simple surfaces. Some parts are somewhat unclear and could benefit from further clarification.**

**Originality:** 3
**Technical Quality:** 3
**Clarity Of Presentation:** 3
**Potential Impact:** 3
**Recommendation:** 3
**Confidence:** 4

**Review:**

and exploration of this field.

Major Concerns and Questions:

- The benefit of adopting a quadric function to correct depth noise for environments with simple surfaces is clear. However, have you tested the method on scenes where the object’s surface is not locally smooth, such as a durian or some other objects? Including this additional environment could provide readers with a better understanding of the proposed quadric representation's capabilities.

- The scene is initially divided into millions of cubes. Is this division solely for obtaining quadric surfaces? In other words, do you use a single MLP for the entire scene representation, similar to NeRF?

- Line 107 mentions that the tracking module predicts rough depth maps and initial camera poses. This was confusing until I read Section 4.1, which clarifies that the pre-trained DROID-SLAM is used as the tracking module. It would be helpful to mention this with proper citation around line 107 to make the paper easier to understand.

- What pre-trained segmentation network is used for obtaining the segmentation map? Please clarify this with proper citations. Additionally, how many labels does the segmentation map provide? Is it trained to identify flat or nearly-flat surfaces rather than objects, as shown in Figure 2? Providing more details would be beneficial.

- Regarding quadric depth correction, are the parameters estimated per depth? In other words, do you calculate the quadric parameters in the world frame and transform them into the camera local frame using the camera pose to get the depth map, or do you estimate the parameters for each local frame? It makes sense if the parameters are in the world frame, but based on Section 3.2, it seems they are estimated in the latter way.

- For sampling points, do you sample points along the ray, like NeRF, or do you use the depth map as a guide? Please provide more details.

- For joint optimization (lines 184-187), do you refine a single camera pose or jointly estimate the camera poses of multiple frames? The former method is more akin to visual odometry rather than SLAM.

- In Table 2, why is the performance of the proposed method on datasets 0059-0181 not as good as on the others? I am curious about what kind of environments or trajectories affect the performance. Even a preliminary explanation would be helpful.

**Quality Of The Limitations Section:**

2

**Questions For Rebuttal:**

Please refer to my review comments above.

**Robotics Focus:**

3

**Summary Of Paper:**

This paper proposes a framework that combines volume rendering with SLAM. Instead of representing the entire scene using an MLP like NeRF, the proposed method divides the space into millions of cubes and extracts quadric planes for environment representation. Specifically, raw RGB images are input into the pre-trained DROID-SLAM, which provides initial poses and depth maps. The RGB images are also fed into a pre-trained segmentation network to obtain segmentation masks. The predicted depth maps and segmentation masks are then fused under the quadric assumption to produce a refined depth map. Utilizing the refined depth map, the camera poses and NeRF parameters are optimized, with a transformer used to share information between sampled points along and between the rays.

**Summary Of Recommendation:**

This paper enhances DROID-SLAM by incorporating quadric depth correction and the ray transformer. - Some parts of the paper are not very clear, and I would like to see the authors' response to these points. - The attached code is helpful in demonstrating and validating the proposed method.

---

### Official Review · Reviewer_V7cK · 2024-07-23
**Good 3D reconstruction work with many concerns and details to be addressed. Additionally, it should not be presented as a SLAM Paper.**

**Originality:** 2
**Technical Quality:** 2
**Clarity Of Presentation:** 2
**Potential Impact:** 2
**Recommendation:** 1
**Confidence:** 5

**Review:**

**Strengths:**
1. The proposed method shows good results in 3D reconstruction and pose estimation, with comparable memory consumption and latency.
2. Overall, the paper is easy to follow, and the method has been tested in different environments, showing good generalization.

**Weaknesses:**
1. First of all, I strongly oppose the presentation of this paper as a SLAM work. The paper does not mention any of its pose estimation components: how its front-end tracking works, how it handles temporal data, or how the loop closure is implemented (if it claims to be a SLAM paper, instead of an odometry paper). I feel this paper mainly (if not only) focuses on the 3D reconstruction given posed RGB inputs.
2. Incomplete reference work: the related work mainly includes very recent neural implicit RGB-D SLAM, ignoring the conventional works on RGB-D SLAM and Scene Representations.
3. Too many details are missing in the paper. The initial depth prediction/estimation, segmentation, bundle adjustment, and camera tracking components are barely mentioned. Without these details, it is impossible to reproduce or even judge the quality of this work. These are also essential components of a SLAM system. If the pose estimation and optimization stages are not explained, there is no point in evaluating them.
4. There is no discussion of limitations or failure cases.

**Quality Of The Limitations Section:**

1

**Questions For Rebuttal:**

* The experiment hardware setup for training and the inference speed have been mentioned in the paper. Is the test platform for inference speed the same as the training platform? Referring to Table 3, on which test platform were the experiments conducted? Were the three methods all tested on the A6000 GPU? Additionally, there should be an analysis of its components (refer to Figure 6 in the appendix).

* How does this work compare to other quadric-based SLAM methods, especially the following:
  * Laidlow, Tristan, and Andrew J. Davison. "Simultaneous localisation and mapping with quadric surfaces." 2022 International Conference on 3D Vision (3DV). IEEE, 2022.

* Table 3: All the comparison methods require heavy GPU computation, which is impractical for some power-limited robot platforms. Can the authors also please compare some dense SLAM methods that have been widely adopted on mobile robot platforms:
  * Vespa, Emanuele, et al. "Efficient octree-based volumetric SLAM supporting signed-distance and occupancy mapping." IEEE Robotics and Automation Letters 3.2 (2018): 1144-1151.
  * Oleynikova, Helen, et al. "Voxblox: Incremental 3D Euclidean signed distance fields for on-board MAV planning." 2017 IEEE/RSJ International Conference on Intelligent Robots and Systems (IROS). IEEE, 2017.

* There needs to be a justification for why the optimized NeRF field needs an extra post-processing step to run TSDF-Fusion again. The NeRF field can be directly converted to 3D meshes.

* I am also a bit confused about Table 5 in the appendix. It reports that ORB-SLAM2 and ORB-SLAM3 failed on half of the TUM RGB-D datasets. This is confusing as I have personally tested ORB-SLAM on these scenes, and it definitely did not show such a high failure rate.

* How does the method scale to larger environments? Since it is based on quadric surface representation, I assume it should have better memory efficiency compared to volumetric representation. But this needs to be confirmed in the experiment.

* The quadric ray sampling adds some regularization, implying that points on the same quadric exhibit similar features. Will the quadric representation cause some loss of geometric details? Is it sensitive to segmentation errors? For example, in the ScanNet scene where objects are clustered, objects tend to be missed in the segmentation. Will the semantic error affect the geometry quality?

* Some notations in Section 3.3 are not explained.

**Robotics Focus:**

3

**Summary Of Paper:**

This paper presents a novel 3D reconstruction method using compact quadric representation with neural implicit representation.

**Summary Of Recommendation:**

This work shows some advancements in 3D reconstruction, but too many details are missing for it to be considered a SLAM paper. I would strongly reject it as a SLAM paper unless the authors are willing to reformat it as a 3D reconstruction work. Maybe it is more suitable in a computer vision venue as a 3D vision paper.

---

### Official Review · Reviewer_Z7x8 · 2024-07-24
**Q-SLAM: Quadric Representations for Monocular SLAM**

**Originality:** 3
**Technical Quality:** 4
**Clarity Of Presentation:** 4
**Potential Impact:** 3
**Recommendation:** 4
**Confidence:** 3

**Review:**

This paper shows how to impose quadric-constraint on scene segment in the context of 3D SfM (or SLAM). The core idea is to use an image segment (from a segmented network), fit a quadric to them and force the segment points ot the quadric. The corrected depth can then be used to support scene reconstruction with a NERF. A second contribution is to use a ray transformer to improve NERF performance.

The use of quadrics to approximate parts of the scene is very interesting. In a way the quadrics are used to regularize the depth. This regularization is claimed to improve the precision of the depth reconstruction but this is not specifically evaluated. Another point that would be worth discussion is the sensitivity of the fit to outliers: would it be helpful to run a RANSAC quadric fitting per segment, before projecting the segment points on the surface, or is it not necessary because NERF will handle it later ?
When looking at the video, the resulting depth seems to be extremely smooth. This is a property of the scene but may not be true everywhere, specifically in scene segment containing vegetation for instance.

The second contribution is the use of a specific ray-transformer to improve NERF performance. The transformer itself seems to be independent of the quadric representation, but the papers shows that the quadric prior supports the sampling of the rays.

Overall, the evaluation and comparisons with sota are by the book, with the tables as expected. An ablation study is provided but not a specific limitation section.

Although the approach is extremely interesting, the global impression is that the paper is trying to fit too much in the limited space, which results in a limited amount of details and insight.

**Quality Of The Limitations Section:**

1

**Questions For Rebuttal:**

The main questions to address would be the discussion on the robustness to outliers, and the dependence between the ray transformer model and the quadric sampling for NERF. Could these two components be evaluated independently in the ablation study?

**Robotics Focus:**

3

**Summary Of Paper:**

This paper shows how to impose quadric-constraint on scene segment in the context of 3D SfM (or SLAM). The core idea is to use an image segment (from a segmented network), fit a quadric to them and force the segment points ot the quadric. The corrected depth can then be used to support scene reconstruction with a NERF. A second contribution is to use a ray transformer to improve NERF performance.

**Summary Of Recommendation:**

This paper proposes an interesting integration of priors in the Nerf-based SfM pipeline. This is relevant to robotics but mostly a computer vision contribution.

---

### Author Rebuttal · Authors · 2024-08-12

Dear Reviewers **V7cK**, **csiv**, **Z7x8**,   and Area Chair **yjt5**,


We sincerely appreciate your thorough review of our paper and the valuable feedback you have provided. We would be grateful if you could review our rebuttal and share any additional feedback or questions you might have.

As the rebuttal period is ending soon, we are wondering if the rebuttal clarifies any points of confusion. We have provided a revision with changes highlighted in blue text.

Thank you once again for your time and efforts!

Best,

Authors.

---

### Decision · Program_Chairs · 2024-09-04

**Decision:**

Accept

**Comment:**

Although the paper is recommended for acceptance, it is strongly encouraged that the authors consider changing the presentation to a paper that focuses on 3D reconstruction on the camera-ready version (including the title).

Note that the novelty of the quadric representation is clear for 3D reconstruction, while the SLAM part uses existing approaches.

For the rest, the authors successfully addressed most of the reviewers' concerns. Please include all the required modifications in the camera-ready version.

Summary of Strengths:
* Interesting integration of quadrics
* Good results compared with the state-of-the-art
* Quadrics are used to regularize the depth

Summary of Weaknesses:
* Limitations are not well addressed
* The paper is about the scene representation not really about SLAM
* Clarifications and more details are needed
* Missing references
* Missing evaluation of quadrics are used to regularize the depth